# Strategies for Reducing Sodium Intake in Bakery Products, a Review

**Georgiana Gabriela Codină *** , **Andreea Voinea and Adriana Dabija**

Faculty of Food Engineering, Stefan cel Mare University of Suceava, 720229 Suceava, Romania;
andreea.musu@fia.usv.ro (A.V.); adriana.dabija@fia.usv.ro (A.D.)
* Correspondence: codina@fia.usv.ro or codinageorgiana@yahoo.com; Tel.: +40-745-460-727

**Abstract:** Nowadays, the dietary sodium chloride intake is higher than the daily recommended levels, especially due to its prominent presence in food products. This may cause an increase of high blood pressure leading to cardiovascular diseases. Cereal products, and in particular bread, are the main source of salt in human diet. However, salt is a critical ingredient in bread making, and its reduction can have a negative impact on bread quality. This review focuses on physiological role of sodium chloride, its effect on the human body and legislative recommendations on its consumption. Moreover, it presents sodium chloride effects on the bread making from the technological and sensory point of view and presents different options for salt reduction in foods focusing on bakery products. It may be concluded that salt reduction in bread making while maintaining dough rheological properties, yeast fermentation rate, bread quality through its loaf volume, color, textural properties, sensory characteristics is difficult to be achieved due to sodium chloride's multifunctional role in the bread-making process. Several strategies have been discussed, focusing on sodium chloride replacement with other type of salts, dry sourdough and flavor enhancers.

**Keywords:** salt reduction; legislative recommendations; bread making; salt replacement; bread quality





## 1. Introduction

Bread is a part of the foodstuffs that are the basis of many people's diets due to its nutritional value and the low price that is reflected from the flour from which it is obtained, the auxiliary materials used, and the technology applied. Food experts define bread as a staple food at the top of the food pyramid due to its rich content in carbohydrates, fiber, protein, B vitamins and mineral salts [1]. In most European countries bread is the most important sources of salt, its contribution to salt intake ranging between 19.1% in Spain to 28% in France [2].

According to the World Health Organization, processed foods such as bakery products are the main daily source of sodium in consumers' diet for developed countries, with an average of about 75–85% of the total sodium intake, while 5–10% are naturally provided from foods consumption that are part of the daily diet and the remaining part of 10–15% of sodium are provided from sodium chloride addition during cooking or eating [3]. However, in developing countries, salt addition during meal preparation presents a much more important role [4]. Globally, the sodium intake from processed foods is much higher than the intake of unprocessed, naturally consumed foods [5].

Although sodium is a normal constituent of the human body, distributed in the extracellular compartments, performing many functions with beneficial effects on the body, excessive sodium intake is associated with cardiovascular diseases caused by increased hypertension [6].

Epidemiological studies on hypertension have shown that many people from countries where salt consumption is high also presented high hypertension values [7]. Often, the salt consumption covers 35–50 times the renal needs and it can be concluded that one of the fundamental characteristics of contemporary diet is an excessive salt intake. It is considered

that two main factors have contributed to this situation: a behavioral factor in humans salt intake which is not dependent on real needs but on the taste for salt artificially created and which is part of a hedonic behavior that develops since childhood; the second factor is the urbanization that has led to a consumption of industrialized foods in which sodium chloride is used as a flavor additive and preservative. As a result, increased consumption of salt is a relatively recent food habit, which greatly demands the body's ability to adapt [8].

Nowadays, consumers' concerns about excessive sodium intake and its associated effects have increased, and that is why some food companies have changed their product portfolios to reduce sodium intake and to promote healthier diets. An example of this is Nestle, which eliminated almost 7500 tons of sodium from their products starting 2005 [5].

Although consumers are now becoming aware of the negative effects of salt excess consumption on their health, they do not have much information on the salt connections with sodium consumption. In developed countries, consumer awareness of proper nutrition and nutritionally healthy behavior is increasing nowadays especially through education [9,10]. Unfortunately, in underdeveloped countries the level of education regarding proper nutrition behavior is very low and therefore the population awareness on the negative effect of excessive sodium consumption on health is not very high [10].

From the consumer behavior point of view, sodium chloride increases the acceptability of many foods by intensifying the salty taste and flavor and by trans-modal interactions which increase the taste of other aromatic compounds and decrease or eliminate the bitter taste [3]. It seems that the sensitivity to salt varies in the same individual from one moment of life to another: depending on age, blood pressure level, obesity, pregnancy, various diseases, drugs consumed, and even race etc. risk [11].

Sodium chloride is one of the raw materials in the bakery industry, which is used to make all bakery products except salt-free dietary products. It has an important role on the sensory characteristics of bakery products but also on the technological characteristics of bread making such as dough rheological properties, enzymatic and microbiological dough activity, and bread quality [11]. Nowadays, bread is considered one of the most important sources of salt in the diet, contributing 25% of the amount of salt consumed by the population. Therefore, for a reduction in salt intake it is necessary a reduction of salt in the bakery products. In general salt is used as a food ingredient, as a preservative, to improve moisture retention and to increase food sensory characteristics. Although in some cases it is impossible to reduce the salt content from foods, in many others it is possible to obtain processed foods with lower sodium content. This is also the case for bakery products which are the largest contributor to dietary sodium intake in Great Britain and the USA [12].

Nowadays, almost every EU country has different strategies which includes recommendation for salt reduction via food reformulation to reduce the salt content from food products including bakery ones. For example, different programs are developed in EU such as "STOP SALT!" in Hungary, "Gaining health: making healthy choices easier" in Italy which encourages in especially salt reduction in bread, in Bulgaria the National Food and Nutrition Action Plan 2012–2017 promotes salt reduction, etc. The bakery products reformulation strategies for salt reduction are continuing in EU countries, some examples in this regard being the following: in Austria the salt reduction has been established of 15% up to 2015 by the Federal Ministry for Health, in Italy of 10% up to 2012 by the Ministry of Health, in Spain of 20% up to 2014 by the Ministry of Health and Social Policy, in Hungary, the Hungarian Bakery Association recommended reducing salt in bread to reach, after December 2018, a maximum level of 2.35%, etc. [13].

Numerous strategies have been proposed to reduce sodium chloride in foodstuffs including bakery products, in order to improve the health of the population. The challenge of these strategies is to solve the technological and sensory problems caused by sodium chloride removal from bakery products recipe. From the technological point of view, a salt reduction up to 0.6% may conduct to bakery products without a significant negative effect on dough rheological properties. However, its effect on the sensory properties of bakery

products may be a problem due to the fact that the salty taste is difficult to be achieved [1]. There are a number of combinations of substances proposed as salt substitutes, which will be discussed in a more detailed way in point 4 of this review, but they still may offer to the bakery products an unpleasant taste. Different combinations based on flavor enhancers, other type of salts, only partial substitution of sodium chloride from bakery recipe may offer some solutions for sodium chloride reduction. More, to reduce the negative effects of lack of salt on technological properties of bakery products it is recommended the use of another type of salt with similar effects on dough rheology as the one produced by sodium chloride. The proposed strategies for salt reduction are shown in Figure 1, methods that as we mentioned may be combined in order to increase bakery products flavor and salty taste.

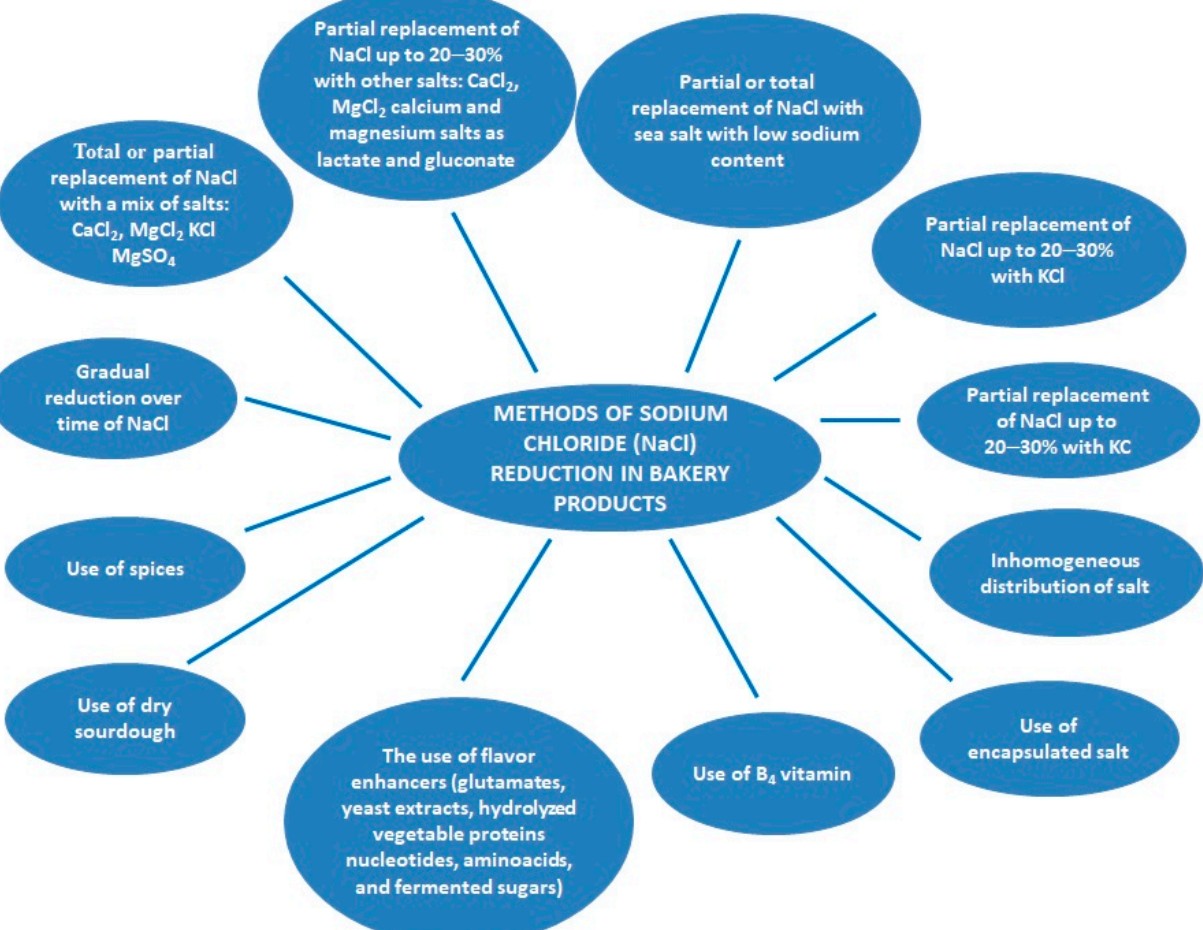

**Figure 1.** Strategies of salt reduction in bakery products.

All these strategies along with sodium chloride effect on bread making, its physiological role on the human body and legislative norms regarding the international recommendation of daily sodium intake will be discussed further.

## 2. The Physiological Role of Sodium Chloride and Legislative Recommendations on Its Consumption

Sodium is among the top six elements in the earth's crust, comprising 2.83% sodium in all its forms. The most important sodium salts found in nature are sodium chloride, sodium carbonate, sodium borate, sodium nitrate and sodium sulfate [14]. To obtain sodium chloride, the weight of sodium must be multiplied by 2.54. Regulating the levels of sodium and chloride in the human body is an important biological process, maintained by multiple mechanisms that work to control them. In the human body, sodium and

chloride are the major constituents of the extracellular fluid, participating in maintaining the electrochemical gradient between the extracellular space and the cytoplasm. In the absence of this gradient, life is not possible. Their presence in the human body maintains the membrane potential that is absolutely necessary in the transmission of nerve impulses, muscle contraction and therefore cardiac function. Absorption of sodium in the small intestine promotes the assimilation of amino acids, glucose and water [1].

Sodium is a colorless crystalline compound found naturally in many foodstuffs and its most widely used form is sodium chloride, also named table salt. Salt has an important role in the human history. The ancient populations used salt to preserve food, salt being for them so precious that it could be exchanged for gold. Until the end of antiquity, salt was, along with amber and tin, one of the main currencies. In American history, salt has been a vital element for survival and during the Civil War it was used not only for foodstuffs but also for tanning the skin, dyeing clothes and preserving rations [15]. The need for permanent salt intake is one of the physiological needs to which man is exposed. Blood sodium concentration is an important homeostatic parameter that controls extracellular fluid volume and blood tonicity [11].

Nowadays salt is used as a spice and preservative in processed foods, being used in various forms such as sodium nitrite, sodium benzoate, monosodium glutamate and baking soda. It is present in processed foods such as soy sauce, canned meat and vegetables, soups, processed meat and almost any food with a long shelf life [15].

### 2.1. Aspects Regarding the Effect of Sodium Chloride Consumption on the Human Body

Sodium ($Na^+$) is the dominant cation in the extracellular fluid of the human body. The functions of sodium consist in its participation in the control of the volume and systemic distribution of the total water in the body allowing cellular absorption of dissolved substances and generating, by interactions with potassium, of the trans membrane electrochemical potential [16]. Sodium ions are involved in transmitting electrochemical impulses along cell membranes to maintain normal nerve and muscle susceptibility. They contribute to the swelling of colloids in the tissues and thus cause the retention of bound water in the body. At the same time, sodium takes an active part in neutralizing the acids that form in the body. It is an element present in all organs, tissues and biological fluids, which plays an important role in intercellular processes and interstitial metabolism [17].

In the body of an adult with a weight of about 70 kg, sodium is found in the body in an amount of 92 g and is distributed differently from person to person but in identical concentrations in all people, regardless of sex, age and physical or intellectual effort [18].

Sodium has multiple roles in the human body, including:

- maintaining a normal excitability of the muscular and nervous system;
- activation of enzymatic systems;
- maintaining a normal pH in the stomach, intestines and blood plasma;
- regulation of water absorption and retention [19].

Sodium is essential for cellular homeostasis and physiological function. Claude Bernard was the first to highlight the "internal environment". Walter Cannon defined homeostasis more clearly when he referred to the "fluid matrix" of the body and emphasized the role of sodium [20]. In the last few decades, there has been an increasing amount of work exploring sodium and dietary health. The amount of sodium needed to maintain homeostasis in adults is extremely low (<500 mg) compared to the average intake of most Americans (>3200 mg) [21].

Dietary sodium deficiency is rare in healthy European populations. Sodium chloride and other sodium salts are daily used in the diet and there are adaptive physiological mechanisms that reduce the loss of sodium from urine, feces and sweat to a low sodium intake. Sodium chloride addition during industrial food processing or food preservation is the main source of dietary sodium in Western diets. Other sources of sodium include inherent native sources and sodium-containing food additives, in which sodium may be associated with anions other than chloride.

In healthy people, almost all dietary sodium is absorbed, even at a high sodium intake. After absorption, sodium ions are distributed through portal and systemic circulations, where their concentrations are maintained in a limited range. Up to 95% of the body's sodium content is in the extracellular fluid, including a large amount in bone, skin and muscle. Sodium excretion and retention (homeostasis) is performed by an integrated neurohormonal control from the centers located in the hypothalamus. The kidney is the main organ that mediates the excretion and retention of sodium. It effectively excretes sodium in response to high sodium intakes from food and stores sodium when dietary intake is deficient. In contrast, the excretion of sodium in the feces is relatively stable and usually limited to a few mmol/day. The amount of sodium excreted through perspiration can vary greatly, depending on environmental conditions or levels of physical activity [16].

Excess sodium in the diet has been linked to high blood pressure. The sensitivity of blood pressure to salt varies greatly, but certain subgroups tend to be more sensitive to salt. The mechanisms underlying sodium-induced increases in blood pressure are not fully understood, but may involve changes in renal function, fluid volume, fluid-regulating hormones, vasculature, cardiac function, and autonomic nervous system. It was established that in hypertension cases excess salt is not only an aggravating factor, but can also be one of the triggers. It appears that there is a functional abnormality of red blood cell membranes in hypertensive population, which it is reported to the ratio between the net flows of $Na^+$ leaving and the net flow of $K^+$ entering. This abnormality is genetically transmitted and allows sodium to enter into the cells in excess. Many arguments support the role of salt as a factor in promoting atherosclerosis. Recent preclinical and clinical data suggest that, even in the absence of an increase in blood pressure, excess dietary sodium can adversely affect target organs, including blood vessels, heart, kidneys, and brain [22]. In normal individuals, the increase in excess salt has no short-term effects. In the long term, if the kidney does not have the ability to regulate the concentration of sodium, as a result of excess of NaCl consumption, there is an increase in blood pressure in the peripheral vessels. To compensate the excess of $Na^+$ and to prevent high blood pressure, a higher amount of natriuretic hormone is secreted. As a result, the sodium pump is injected from the erythrocytes and the smooth muscles of the vessels, which causes the membrane depolarization and the internal accumulation of calcium. Gradually, hypertonia is reached [8].

### 2.2. Legislative Norms on Recommended Daily Sodium Intake

Globally, institutions such as the European Union are actively promoting the reduction of salt content in food. The European Parliament and the EU Council approved in 2006 a regulation on nutrition and healthy food (EC) No. 1924/2006 (European Commission (EC), 2006), this document allowing, among other things, the use of nutritional claims regarding the sodium/sodium chloride content of foods. These regulations allow consumers to focus on the salt content of food with the help of inscriptions found on product packaging [1]. Article 8 of EU Regulation 1924/2006 lists the following restrictions for sodium/sodium chloride claims it's shown in Table 1.

**Table 1.** Nutrition claims regarding salt/sodium content—per Article 8 of EU Regulation 1924/2006 [1,5].

| Salt Content, g/(100 g/100 mL) | Sodium Content, g/(100 g/100 mL) | Nutrition Claims |
|---|---|---|
| - | - | Low content (sodium chloride/sodium) certifies that the sodium or equivalent salt has been reduced by at least 25% compared to a similar product. |
| 0.30 | 0.12 | Low sodium chloride/sodium content certify that the product does not contain more than 0.12 g of sodium or the equivalent value for sodium chloride, 0.3 g per 100 g or per 100 mL. |

**Table 1.** *Cont.*

| Salt Content, g/(100 g/100 mL) | Sodium Content, g/(100 g/100 mL) | Nutrition Claims |
|---|---|---|
| 0.10 | 0.04 | A very low sodium/sodium chloride content certifies that the product does not contain more than 0.04 g of sodium or the equivalent value for sodium chloride, 0.1 g per 100 g or per 100 mL. |
| 0.013 | 0.005 | Sodium-free or sodium chloride-free certifies that the product does not contain more than 0.005 g of sodium or the equivalent value for sodium chloride, 0.013 g per 100 g. |

*2.3. Sodium Chloride Content on Bakery Products and Their Degree of Consumption*

The main source of salt in food in most European countries is generally foodstuffs such as bakery products, followed by meat and meat products, cheese and dairy products [23–25]. Bakery products have a major contribution to the daily intake of sodium in the diet, along with other products obtained from cereals such as biscuits, cakes, breakfast cereals, pastries, noodles, cereal bars, etc. The origin of sodium in these products is caused by the addition of sodium chloride to obtain them but also other raw materials and ingredients used. For example, a source of sodium is also ingredients such as baking powder or leavening agents [26]. It was found that the daily consumption of 150 g of bread containing 20 g of salt/kg of flour contributes with 25% of the average amount of salt consumed, which is 10 g/day/person. Bread is thus the main food contributor to the sodium chloride intake in the diet. Per capita, bread consumption varied between EU countries. The highest bread consumption was reported in Turkey (104 kg per year) and Bulgaria (95 kg) while the lowest one was reported in Great Britain (32 kg). It seems that the average consumption of bread by European people are 59 kg per year. Bread consumption is generally stable but in some countries such as Netherlands, Belgium, UK, Poland has been reported a slightly decrease. In 2015, in Poland, the bread consumption was of 145 g of bread daily. Thus, consumption of 145 g of wheat baguettes provides 4.4 g of salt per day, covering 87% of the maximum daily salt requirement given by the WHO. However, other types of bread consumption provide 1.6–2.2 g of salt, covering 32–44% of the demand [2]. According to the studies made on several European countries it was concluded that in Ireland bread accounts for 25.9% of total salt intake, in Turkey 25.5%, in Belgium 24.8%, in France 24.2%, in Spain 19.1% and in the UK 19% [12]. Compared to other European countries, in Romania, bread represents 30% of the total salt intake, which indicates a higher value compared to other European countries [27]. According to consumer taste tests, it was reported that the optimal salt content for white wheat flour bread was between 1.29% and 1.43%. In contrast, a similar study in Argentina reported a value of 1.74%, which was much higher than expected and could be due to some geographical preferences. There are major differences in the salt content of different types of bread. In France, for example, bread contains an average of 1.7% NaCl. This amount is found in bread, country style, French baguette whereas in croissant, puff pastry and sweet bakery products the amount of salt is lower of an average of 1.3 and 1% respectively [1]. In the UK bread has only 1.0% salt [12] whether it is white or brown whereas in the fruit buns, plain cake and fruited cake the amount of salt is lower between 0.32–0.72%. In Ireland, the amount of salt in bread is 1.10% in white bread and 1.09% in brown bread and in Germany the amount of salt in all types of bread and rolls is between 1.0–2.9% [1]. In Italy, the salt content of bread varies from 0.7 to 2.3% for artisanal bread and from 1.1 to 2.2% for bread produced to the industrial level [28] and in Spain all types of bread have an amount of 1.63% salt [1]. Thus, the bread contribution to salt consumption highly differs depending on the type and location. In Romania, the salt content of bread varies depending on its type from 0.17% for non-salt bread and 1.79% for potato bread. An important factor in the salt variation of bread is its type. Thus, for example, baguette bread has a lower salt content and potato bread a higher salt content.

The most consumed bread in Romania is the one obtained from refined wheat flour, with a low selling price. This type of bread has an average salt content of 1.25% [27].

## 3. The Technological Effect of Sodium Chloride in Bread Making

Bread is one of the oldest foods consumed in the world and is obtained by baking a dough, prepared from wheat or rye flour, possibly mixed with other legumes or potatoes flours [29]. In bakery products, the salt is introduced into the dough phase in the form of saturated or concentrated solutions, but also in an undissolved state. This has a major impact on the dough rheological properties but also on the finished product quality [24].

The dough is a heterogeneous system which contains carbohydrates, proteins, lipids, mineral salts, water, and air in different proportions, with different degrees of homogenization, being from a rheological point of view a viscoelastic mass and from a technological point a view a homogeneous semi-finished product. It is obtained by mixing flour with water, salt, with or without baking yeast and other auxiliary materials [29]. The dough rheological properties present an important role in the bread making process, being closely correlated with the quality characteristics of the finished product [24,30].

The most important stages in the bread making process are dough making, its processing and dough baking to obtain the finished bakery product [31]. The NaCl effect in bread making process is shown in Figure 2.

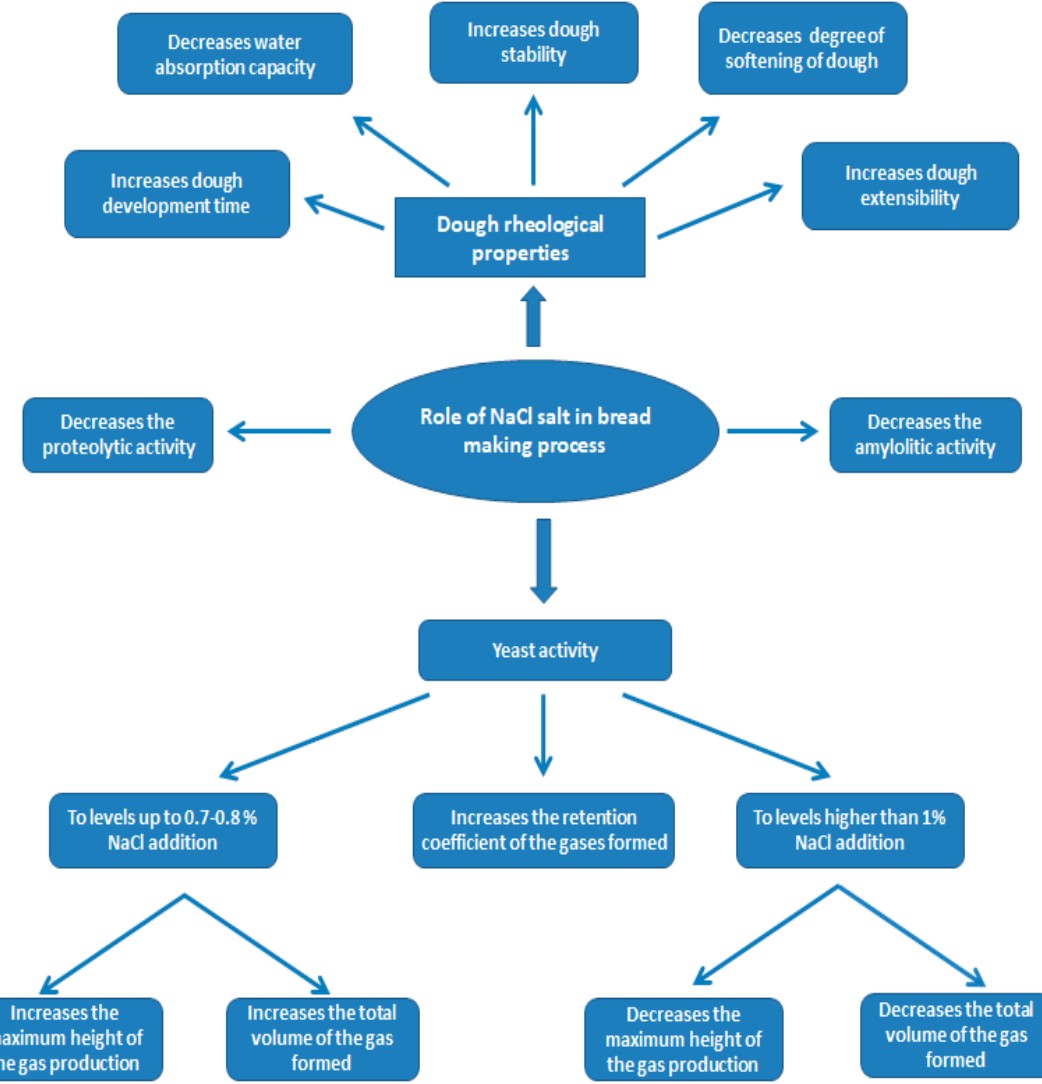

**Figure 2.** The role of sodium chloride in bread making process.

As it may be seen, salt, due to its ionic nature has significant influences in the dough making such as dough development, water absorption capacity, mixing time, its intensity, etc. [32–35].

Many parameters such as temperature, mixing time and the ratio between the amount of water and flour are essential in order to obtain optimal dough rheological properties [36]. The addition of salt decreases the flour water absorption capacity and increases the dough development time and its stability. The increase in dough development time shows that salt delays the formation of gluten during mixing, therefore extending the mixing time [37]. Consequently, the more salt is added, the higher the dough development time and the lower the dough softening is. Studies have shown that the dough consistency is lower as the salt addition is higher if the amount of water added to the dough making remains constant. Many researchers concluded that the effect of salt in the dough system is primarily related to the change in the gluten proteins hydration which changes the ratio between free water and bound water, in the sense of increasing the amount of free water. The addition of 2% salt decreases the hydration capacity of gluten by 8% without changing the hydration capacity of starch [38]. This was attributed to the conformational changes from gluten proteins which occur in the presence of salt. The conformational changes may occur due to the salt ions interaction with the electrically charged groups from the protein molecules. As a result, the intermolecular and intramolecular electrostatic repulsion forces between protein molecules are reduced, these becoming more compact. It was suggested that salt, for a wheat dough pH value of 5.6–6, decreases the intermolecular repulsion to a higher extent than the intra-molecular one. The results of these conformational changes of the protein molecules may lead to a decrease of the proteins ability to bind water, as they become more compact and less penetrable to water molecules. Moreover, the fewer hydrophilic groups are more available to interact with water. Additionally, some hydrophobic groups may be greater exposed, leading to more hydrophobic intermolecular interactions between protein molecules which become more compact for water and more resistant to the action of enzymes.

The increase in the amount of free water can also be attributed to the increase in inter-micellar osmotic pressure (external to the protein micelle) following the salt dissolution in the free water from the dough system. As a result, there is a difference between internal and external osmotic pressure and in order to balance the osmotic pressure some of the initial water bounded to the gluten proteins diffuses outside, making them more compact and more resistant [39].

Salt increases the dough extensibility and its strength. This increase depends on the type of cation it includes [38]. Reducing the salt content from the bread recipe mainly affects dough elasticity, without impacting viscosity. Sodium and chloride ions compete for water from the dough system affecting the hydration of the wheat flour proteins. They cannot hold water for a longer period of time, which causes an increase in the amount of free water, changing dough rheological properties [40]. During fermentation, the salt may have an inhibitory effect depending on the yeast strain and its concentration in the dough system. Thus, for concentrations below 1.5% in relation to wheat flour, the inhibitory effect is less, but it will increase with higher salt concentration, due to the rise of osmotic pressure from the dough system. Studies have shown that salt mostly inhibits the maltose fermentation and, in a lower extent, the glucose, fructose and sucrose fermentation.

Some studies reported that the surge of salt addition in dough system led to a significant reduction in the maximum height of the dough, which causes an increase in the total volume of the gas released. Through salt addition the gluten network becomes stronger. Therefore, the gas capacity to retain gases grows as the amount of salt used in the dough recipe increases [41].

On the finished bakery products, the addition of sodium chloride has an important role on their technological and sensory properties as it may be seen from Figure 3. From a technological point of view, it was concluded that the bread with 0.3% and 0.6% sodium chloride addition in wheat flour does not present significant differences compared to the

bread recipe with 1.2% sodium chloride regarding its loaf volume, moisture and losses during baking. However, the lack of sodium chloride causes significant changes in the structure of the bread crumb and the shelf life after five days of storage. Regarding the flavor, crust formation and shelf life of low-sodium bread, important changes have been reported [35,42,43].

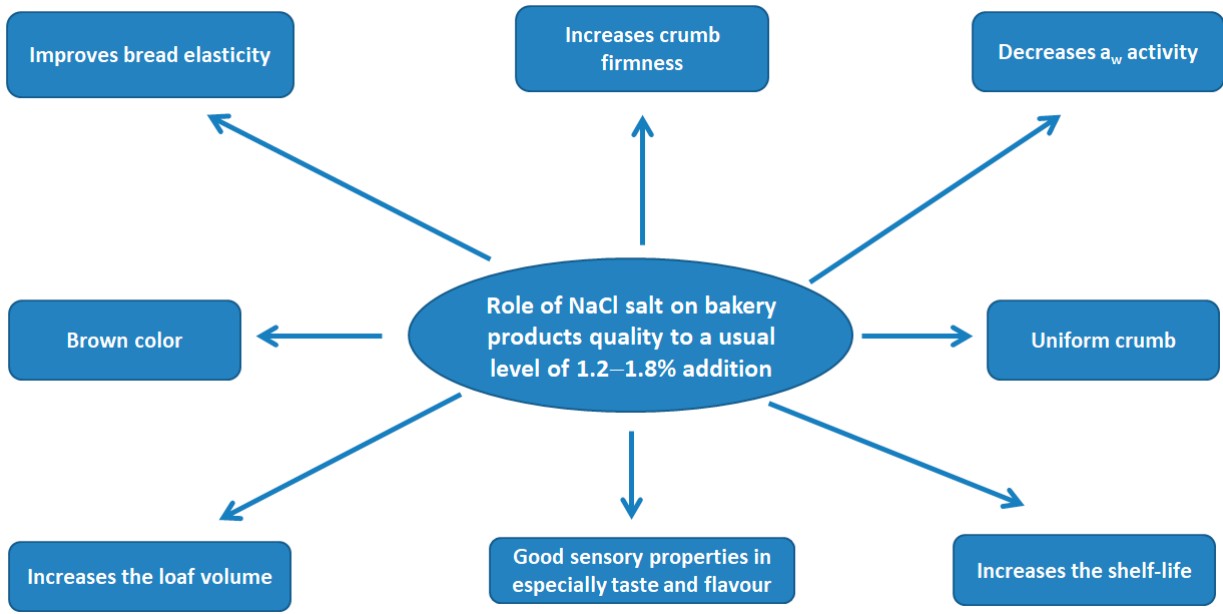

**Figure 3.** The role of sodium chloride on bakery products quality.

The reduction of sodium chloride content in bread influences the textural properties (due to the change in the dough rheological properties), its color and flavor (due to the more intense fermentation activity leading to a reduction in the amount of free reducing sugars resulted from the Maillard reaction) [24].

In bakery products, salt, in addition to its role on the flavor of the finished products, also acts as a preservative. The shelf life of the bread is closely related to its moisture content because it has been found that the migration of water from the crumb to the crust is related to the amount of salt present in the bread [35]. The shelf-life process of bread is due to the starch retrogradation [44], a process which means the tendency of its macromolecular components, amylose and amylopectin, to aggregate, to associate, resulting into a more insoluble form. Degradation is accompanied by the reappearance of crystalline areas into the starch structure partially destroyed during baking, which causes a more rigid crumb. Proteins also contribute to this structure, losing their elasticity and hydration. The water content and its condition play an important role in this process. Thus, products with a moisture content below 16% do not stale. The optimum humidity for staling is between 16–37%. In staling bread, bound water reaches 70% of the total water. The free water from the gel passes into crystalline structures, being strongly bound. It disappears from the bonding layer between the gluten and the partially gelatinized starch granules, causing stronger gluten-starch bonds, which reduces the elasticity of the structure [45]. Salt prevents water migration from crumb to crust and keeps the product fresh for a longer period of time compared to bread without salt addition [46]. The main responsible for foodstuffs' deterioration with high humidity content are microorganisms. Bread is known as a product with high humidity, with $a_w$ values between 0.96–0.98 [47]. In general, the addition of salt increases the gelatinization temperature of starch and delays its gelatinization process [39]. Through its action on starch and gluten, but also on the activity of water from the dough system, salt has a major contribution on starch staling [48]. For example, reducing NaCl from 1.2% to 0.3% decreases the shelf life of the bakery products with almost two days [24].

On the bread color, the reduction of the sodium chloride content in bread recipe or its lack thereof leads to products with a light-colored crust [31]. The brown color of the crust resulting from the bread baking process is due to the Maillard reaction, as an interaction between reducing sugars and amino acids. Therefore, a certain amount of free reducing sugars and amino acids is necessary for this reaction to take place. The lack of salt addition leads to a more intense fermentation which induces a high consumption of carbohydrates during the initial stages of the bread making process. Therefore, during baking the dough which presents less reduced sugars to form more melanoidins, will yield a bread sample with a light color. Many factors such as pH, protein/amino acid ratio, water, sugar amount from the dough system, temperature and baking time have a major influence on crust formation. The intake of sodium chloride produces a plasticizing effect during the baking process of bakery products, this favoring the Maillard reaction and therefore producing a darker crust [49]. Salt has the ability to control the yeast fermentation from the dough system and therefore, reducing the amount of salt will increase the activity of yeast. Fermentable carbohydrates are needed to the yeast fermentation process [50] and therefore, yeast fermentation will reduce the availability of free sugars involved in the Maillard reaction [24].

Sodium chloride is also responsible for the flavor of bakery products, which also intensifies the sweet taste of the finished product. Sodium salts such as sodium chloride or monosodium glutamate influences the food flavor in a positive way, improving it. However, Girgis et al. [51] concluded that consumers did not observe a gradual reduction of 5% sodium per week over six weeks (corresponding to a final reduction of 25% salt). A 20% reduction of sodium in bread did not affect the taste perception of 60 participants in this study. Instead, Lynch et al. [35] showed that a 50% reduction in salt changed the bread flavor. Salt-free bread has been described by them as sour and having the taste of yeast [12]. Research has shown that sodium, and to a lesser extent lithium, are the only salted cations with good effects on sensory food properties while calcium and potassium ions have a bitter or metallic taste, which is undesirable for consumers [52].

This effect of sodium chloride on sensory perceptions are due to the fact that free $Na^+$ ions in solution are the mainly responsible for the perception of salty taste on the tongue while the contribution of $Cl^-$ ion in terms of taste is not yet fully clarified. Due to the interaction between sodium chloride and wheat proteins during dough formation, sodium ions interconnect with dough components due to their ionic interactions with negatively charged amino acids [1]. Pflaum et al. [53] reported that $Na^+$ and $Cl^-$ ions are not irreversibly bounded in bread, sodium ions being released by chewing. The speed with which sodium is released in the first seconds when chewing takes place is quite important for the perception of salty taste by the consumers. The rate of release of the salty taste during chewing depends on the structure of the bread crumb and the amount of NaCl in the product. Bread with low amounts of NaCl has been described as having a more acidic or yeasty taste. Moreover, significant changes in crumb structure have been reported when different amounts of NaCl were added [12].

It has been shown that the use of sodium as compound in different ingredients significantly reduces the perception of the bitter taste [54]. It seems that the positive effects of sodium chloride on the aroma are due to its effect on the water activity. Reducing the amount of free water by adding salt changes the ratio between free water and the bounded water, which influences the volatility of flavor compounds [55]. Moreover, an essential contribution in the formation of flavor compounds is due to the Maillard reaction [56]. As mentioned above, melanoidins are also responsible for the golden-brown color of the bread crust during baking and therefore unsalted or low-salt bread will have a light crust, a weaker aroma, and a bland taste [24]. Obtaining bread with a low salt content (up to 0.3%) is technically easy to be done via a few changes in the bread making process, with a finished product maintaining the same qualities, except for taste. An interesting technique in this respect, although not one that can be applied to all production systems, involves the

inhomogeneous distribution of salt in the bread, which allows a 28% reduction of the salt content, while maintaining the intensity of the salty taste [12].

The textural properties of bread are very important in terms of their acceptance and choice by consumers. They can be evaluated both sensorially and instrumentally. Salt, as mentioned before, has a major impact on the gluten network development during the dough mixing process. The addition of sodium chloride leads to a strengthening effect on the gluten network, stabilizes the fermentation rate of the yeast, increases the dough mixing time, and improves the aroma and porosity of the bread crumb [57].

During the dough making and baking, the appearance of gas bubbles leads to the dough expansion and finally, the loaf volume and texture of the finished product. Gas bubbles have a limited expansion correlated with their stability and possible gases released from the dough system. Therefore, it is very important that the walls of the gas bubbles to be stable during baking but also during other technological stages. Moreover, the dough must present a good gas retention capacity for porosity, elasticity and a specific loaf volume in order to be accepted by consumers [1].

As the salt content of the dough decreases, the activity of the yeast increases leading to a finished product with a higher loaf volume. At low salt amounts levels of 0.7–0.8% the yeast cells multiplication increases, but above this concentration the yeast multiplication process is slowed down due to the plasmolysis process from the yeast cells. Moreover, the addition of salt in the dough reduces the activity of enzymes, both proteolytic and amylolytic, due to the action of salt on the protein part of enzymes [45]. However, less salt can induce a weakening effect on the protein network and to a lower gas retention capacity which will lead to bread with irregular porosity and low loaf volume [31].

From a textural point of view, a comparative study between a bread without salt and a bread with a salt content of 4% reported that there was a high difference between bread samples of about 40% in the firmness value of the bread crumb after a 24 h storage period. This shows that the firmness of the bread crumb decreases with the reduction of the amount of salt from bread and is closely related to the increase of the loaf volume of the finished product [58]. The increase in firmness with the presence of salt in bread can be explained by the conformational changes of the gluten proteins in the salt presence [45].

## 4. Methods for Sodium Content Reduction in Bakery Products

With the growing interest of consumers in a healthy diet, their concern for low-salt products has also increased. This is also due to the media regularly sending messages about the diseases caused by excessive consumption of sodium chloride and its impact on the human body. An example of this is the "Salt Kills" campaign, in which the UK government regularly reports on the harmful effects of excessive salt consumption [59].

According to Powles et al. [60], the average global level of sodium consumption in 2010 was approximately 3.95 g/day, with a specific regional intake of 2.18 to 5.51 g/day. This value is almost twice the limit of 2 g sodium/day (equivalent to 5 g/day of salt intake) recommended by the World Health Organization (WHO) [61].

Salt consumption among adults in most European countries varies between 7 and 13 g per day, according to the European Commission. Germany, Cyprus, Bulgaria and Latvia reported the lowest salt intake (6.3–7.3 g/day), while the Czech Republic, Slovenia, Hungary and Portugal reported the highest salt intake (12.3–13.6 g/day). It seems that are different levels of salt intake in Europe, the lowest values being noticed in Denmark, the Netherlands and Belgium (8.3–8.8 g/day) and the highest in Hungary, Slovenia, Slovakia, Portugal and Italy (10.7–11.2 g/day) [5].

It was concluded that from all the risk factors related to our diet, a diet high in sodium accounted for the highest number of deaths from all the analyzed causes. Most people consume more salt than the World Health Organization's maximum daily salt recommendation of <5 g per day, with an average overall salt intake of about 10 g/day. The daily intake of salt in most countries varies between 9–12 g, which is far above the recommendations of the World Health Organization. It is estimated that the overall reduction in sodium intake to the

recommendation levels would prevent approximately 2.5 million deaths annually, which has led WHO Member States to agree to reduce the sodium intake of the population by 30% until 2025 [62]. A decrease in salt intake by 3 g/day would reduce systolic blood pressure by 5 mm Hg at the age of 60, hence a decrease in stroke and myocardial infarction. Many studies in Germany, Canada and France have shown a correlation between left cardiac ventricular mass and salt intake (assessed by urinary sodium excretion). These negative effects of salt appear to be independent of blood pressure. In addition, the role of sodium in osteoporosis has been demonstrated since the 1970s [11]. Since 2013, World Health Organization member states have committed to reduce the population's salt intake by 30% until 2025 [63]. Many countries have initiated salt reduction interventions, and a number of reviews have shown some progress in reducing salt intake among the population. To further support countries in implementing salt reduction interventions and achieving a reduction in salt intake among the population, the World Health Organization published the SHAKE technical package for salt reduction in 2016, consisting of implementation strategies in five key areas, including surveillance, product reformulation through lower salt intake, labeling, consumer education, etc. Continuous monitoring and evaluation of these efforts is essential to strengthen policies and actions to reduce salt consumption.

The main sources of salt are bread, cheese, meat and meat products, snacks, sauces, soups and pastries. Redeveloping these types of foods to reduce salt content (or improving/reformulating foods) is considered a promising strategy for reducing the dietary intake of salt. In order to reduce the salt intake of a certain population, large-scale structural efforts are needed to reduce the salt content of foodstuffs at the time of production, as well as to initiate behavioral changes. The World Health Organization encourages a multisectoral approach, including public-private partnerships to improve the composition of the food supply [12].

However, traditional culinary habits and consumers' preference for salty taste creates difficulty to the initiative to reduce the consumption of salt in food products. That is why research is underway for different solutions to reduce the content of sodium chloride in food products to be accepted by a large mass of consumers [64]. The gradual reduction in sodium levels by various food industry producers has been somewhat successful, which has been reported by the Food and Drink Federation and the British Retail Consortium. The strategy entitled "reduction in small steps" aims to slowly and gradually to reduce the salt content in manufacturing recipes without notifying the consumers [12].

Bread is one of the main sources of salt in the world and that is why some countries are trying to reduce the sodium content of the general consumption of the population by reducing it in bread. For example, in New Zealand, there has been a 7% reduction in salt content in bread products for four years. In Ireland, through a salt reduction program, the bakery industry set 0.45 g of salt per 100 g of bread as the average value for white and dark bread. In 2007, as part of the second national nutrition and health program, a third of bakeries in France reported a reduction in the amount of salt addition in bread. Similarly, in 2005, the Spanish Confederation of Bakeries agreed to reduce the salt content of bread from 22 to 18 g to 100 g, reducing, on average, 1 g/kg of flour per year. Thus, evidence from various countries shows that salt reduction programs in bakery products are efficient, as part of a cost-effective strategy for improving public health [65].

Studies have shown that a lot of effort has been made to reduce sodium in bread. The salt content of bread in the UK decreased by about 20% from 2001 to 2011 (from 1.23 to 0.98 g/100 g). By 2011, 71% of all bakeries met FSA targets of $\leq 1$ g/100 g, while in 2001 only 25% of all bakeries met this limit [5]. In terms of feasibility and marketing, it seems that salt reduction can be achieved in most bakery products [12]. Different strategies of salt reduction in bakery products are shown it Table 2.

**Table 2.** Strategies of salt reduction in bakery products.

| Strategies | Effects | References |
|---|---|---|
| Sodium chloride replacement with different types of salts | Partial replacement of NaCl with KCl up to 20–30% in bakery products without negative effects on its metallic or bitter taste. This is one of the best solutions reported so far to reduce the sodium chloride from bread making | [24,66,67] |
| | Partial replacement of NaCl with a mix of KCl, MgSO$_4$, MgCl$_2$ up to 32% in dark bread conducted to finished products with a similar flavor and texture with that of the control sample | [68] |
| | KCl addition in bread making presented similar effects on dough rheological properties compared to NaCl salt | [69–73] |
| | NaCl replacement with potassium citrate up to 30% conducted to good results from bread quality point of view | [74] |
| | A mix of KCl and potassium bicarbonate of 1:1 conducted to good results for bread quality and also to an improvement of potassium availability for the human body | [74] |
| | NaCl replacement with CaCl$_2$ up to 25% conducted to good results from the rhelogical point of view with a slightly decrease to water absorption capacity value | [69] |
| | NaCl replacement with CaCl$_2$ changed in a differently way dough rheological properties by decreasing dough development time and by increasing water absorption capacity | [75] |
| | The use of sea salt (a natural mix of NaCl, KCl, MgCl$_2$) with a low sodium content conducted to good results from the rheological, technological and sensorial point of view and may lead to a sodium reduction. For the best bread quality results, it is necessary a low amount of sea salt addition of only 0.5%. | [75,76] |
| | The use of calcium salts as gluconate and lactate as replacements for sodium chloride conducted to similar data for the rheological point of view compared with those obtained when NaCl was incorporated in bread making. Their addition conducted to a strengthening effect (more in the case of gluconate than lactate) an increase of dough stability, a decrease of in degree of softening, a positive effect on yeast activity leading to an increase of the gas retention coefficient. However, contrary to the NaCl effect, the calcium salts improved the alpha amylase activity of the dough system. | [77] |
| | The use of magnesium salts as gluconate and lactate as replacers for sodium chloride conducted to dough mixing properties to similar behavior for gluconate salt and a different one for lactate salt (a more weakening effect on dough) compared with those obtained when NaCl was incorporated in bread making. From the yeast activity point of view, it seems that both magnesium salts had a similar effect to that obtained by the sodium chloride addition in wheat flour. | [78,79] |
| Gradual reduction of sodium chloride in bakery products | The sodium chloride reduction up to 25% in six weeks or even up to 52% in four weeks did not affect the consumer's acceptability or bakery products consumption. However, if the reduction of salt in food is done too quickly without necessary adjustments in food taste, the opposite effect can be reached, namely it can lead consumers to other foods rich in salt or to add it during the preparation of meals or to supplement the lack of salt in that food | [51,80–82] |
| Inhomogeneous distribution of salt in the bread recipe | May lead to a reduction up to 20–30% NaCl content from the bakery products with good results from the sensory point of view but may presents negative effects on yeast activity | [83–86] |

**Table 2.** *Cont.*

| Strategies | Effects | References |
|---|---|---|
| Use of encapsulated salt in the bread recipe | The sodium chloride level may be half reduced in the bakery products with similar effects on salty taste on consumers, but this high reduction may present negative effects from the technological point of view on dough development time, on yeast fermentation rate, water activity, bread loaf volume and textural properties. | [87] |
| The use of flavor enhancers in the bread recipe | Intensifies the salty taste perception by using them in combination with NaCl (glutamates, yeast extracts, hydrolyzed vegetable proteins nucleotides, amino acids, and fermented sugars). The use of fermented sugars in addition with sea salt and dry sourdough it may reduce the sodium chloride content up to 0.02%. | [83,88,89] |
| The use of dry sourdough in bread recipe | It improves the salty taste perception, flavor, shelf life and textural properties of the bakery products by using it in combination with NaCl. It may reduce sodium chloride content in bread from 1.5% up 1% with good results from the bakery products quality point of view and in addition with sea salt up to 1.39% for the optimum rheological properties reducing therefore the sodium content from bread by 22%. | [24,90] |
| The use of different spices in the bread recipe | The use of different spices in combination with NaCl presents the disadvantages that may alter the taste and aroma of bakery products and may not be appreciated by everyone. Moreover, a high level of specific spices used might lead to toxicity. In general, this solution is the easiest to be applied in restaurants or when preparing meals at home | [10,88,91–94] |
| The use of $B_4$ vitamin in the bread recipe | It improves the salty taste perception of bakery products and may replace the NaCl up to a level of 25% without affecting the sensory characteristics of the bakery products. | [95,96] |

The difficulties of sodium replacement in bakery products are due in especially to the salty taste produced by it. This taste in produced by the specificity of the Na+ ions on the epithelial channel known as ENaC (epithelial sodium channel). The primary process by which salty taste is detected is due to epithelial sodium channel receptors that respond almost exclusively to sodium ions ($Na^+$). Therefore, the central gustatory system and mesolimbic structures are needed to process the taste signal and hedonic responses to food. A consequence of stimulating the reward pathways of the brain is palatability, which suggests that an individual's preference may be associated with the hedonic neuronal properties of salty foods [97]. Further, are presented in a detailed way the best solutions reported so far to reduce sodium chloride from bread making.

### 4.1. The Use of Different Types of Mineral Salts as Substitutes for Sodium Chloride in Bakery Products

Today, all kinds of combinations of different ingredients are proposed as salt substitutes that do not give an unpleasant taste to the finished product. A natural combination of different salts is the Dead Sea salt (contains NaCl, KCl, $MgCl_2$) which can be successfully used as a substitute for sodium chloride [75]. Natural salt extracted from the Dead Sea has a much lower sodium content compared to normal salt used in food (maximum 7% sodium in the form of sodium chloride) and even more it contains a number of minerals useful to the human body such as magnesium and potassium [98]. Research on the use of low-sodium salt from the Dead Sea in bakery products have been less conducted. This substitution is considered interesting, has potential and may be useful especially when it is used with other strategies, but more studies are needed, given that some changes in bread quality parameters have been reported [75,76]. The most used strategy to reduce the sodium chloride content is its partial replacement with potassium chloride in a proportion level of 20–30% and not higher because it would give the products a bitter and metallic

taste. Thus, a combination of NaCl and KCl may be a viable alternative for reducing the sodium content in the food industry [74]. Other studies have shown that a series of mixtures of various salts represented by chlorides of $K^+$, $Mg^{2+}$, $NH_4^+$ and carbonates can provide a taste similar to that of sodium chloride. Charlton et al. [68] also replaced 32% NaCl with a mixture of potassium chloride, magnesium sulfate and magnesium chloride in dark bread, the finished product obtained having a texture and flavor similar to a bread with only NaCl addition. [12].

However, currently the most widely used method of replacing sodium chloride is with other types of salt. This is due to the fact that sodium chloride also influences bakery products quality from the technological point of view. The lack of salt influenced in a negatively way the dough rheological properties, especially on the final leavening phase, when the dough gas retention capacity decreases. As a result, the crumb elasticity is reduced, and the porosity of salt-free products is not uniform and insufficiently developed. The bread is obtained with a pale and light crust. Technologically, a reduction of the salt up to 0.6–0.3% would be possible without a significant deterioration of the rheological properties or the performance of the yeasts during the bread making process. However, its effect on the sensory properties of bread is still a critical factor in consumer acceptance [12].

Replacing sodium chloride with other types of salt can technologically improve the quality of bakery products. Several types of salt have been used successfully as substitutes for sodium chloride. The basic principle is to replace sodium cations with others or replace chloride anions with anions such as glutamate and phosphate, as a way to give a salty taste. Currently, the easiest option for bread making is to replace NaCl with mineral salts or with other cations such as calcium, magnesium and potassium. The effect of different cations on the rheological properties of the dough is related to their position in the lyotropic series, also known as the Hofmeister series, which classifies ions based on their ability to cause aggregation or dissociation of proteins. Within the series, both anions and cations are classified in the order of the most stabilizing to destabilizing. Stabilizing ions lead to less hydration, more structure and a decrease in protein solubility, while destabilizing ions lead to more hydration and increased protein solubility, thus affecting both hydrophobic interactions and hydrogen binding.

The ranking, starting from the least stabilizing cations is: $NH4^+ > Cs^+ > Rb^+ > K^+ = Na^+ > H^+ > Ca_2^+ > Mg_2^+ > Al_3^+$. Therefore, with the use of a stabilizing cation, it would be expected that the protein-protein interaction to increase and to promote the formation of a stronger gluten network and therefore of a non-sticky dough. Studies have shown that $K^+$ is the best option for maintaining the rheological properties similar to dough with sodium chloride, because $K^+$ is equivalent to $Na^+$ in the lyotropic series. However, this replacement leads to significant challenge of a metallic/bitter taste [99].

### 4.1.1. The Use of Calcium Chloride as a Substitute for Sodium Chloride in Bakery Products and the Benefits of Its Use on the Human Body

Calcium chloride is used in the bakery industry as an anti-caking agent but also as a substitute for sodium chloride. It is a solid inorganic compound at room temperature, soluble in water and its anhydrous salt is hygroscopic. Moreover, calcium chloride can also be a source of calcium for the human body. Given that billions of people suffer from osteopenia and osteoporosis, the addition of calcium chloride as a substitute for sodium chloride in bakery products can be considered appropriate. In the last decade, interest in the effects of calcium on the human body has increased and studies have expanded to include the entire life cycle. A whole range of foods and supplements in which calcium is added are widely used today. Calcium is distributed throughout the body in small amounts and is involved in the processes of vascular contraction, vasodilation, transmission of nerve signals, transmission of intracellular signals and hormonal secretion.

There are three major categories of people at risk for dietary calcium deficiency. These include women (amenorrhea, post-menopause), people with milk allergy or lactose intolerance, and risk groups for poor food intake (adolescents and the elderly) [100].

Calcium is an important element for the health of the human body and has vital functions inside cells by transmitting signals between the plasma membrane and intracellular mechanisms. Extracellular calcium is an essential cofactor in the proper formation of bones. Decreased calcium content in the bones can cause conditions such as osteoporosis, high blood pressure, arteriosclerosis, neurodegenerative diseases, malignancy, degenerative joint diseases. Increased calcium is recommended in colorectal cancer prevention treatments to lower blood pressure and blood cholesterol levels [101]. The uses of calcium chloride in the food industry are multiple as: anti-caking agent, antimicrobial agent, hardening agent, flavor enhancer, moisturizer, nutritional supplement, pH control agent, stabilizer and thickening agent, to improve textural properties, and so on.

Various studies in the bakery products on the total substitution of NaCl with calcium chloride ($CaCl_2$) have shown that this has led to a significant increase in water absorption capacity and a higher degree of dough softening due to the fact that a certain amount of water remains unabsorbed from the dough system. However, it was found that at lower levels (~25%) of NaCl substitution, the water absorption capacity decreased slightly, which a positive fact is because more protein-protein interactions occurred, leading to a stronger gluten network and stronger dough through cohesive forces [69,102].

### 4.1.2. The Use of Magnesium Chloride as a Substitute for Sodium Chloride in Bakery Products and the Benefits of Its Use on the Human Body

Magnesium chloride ($MgCl_2$) is one of the most common water-soluble natural magnesium compounds. It is found naturally in the salt waters of lakes or in sea water. Magnesium chloride is obtained from mineral salt deposits after the extraction of potassium chloride but also by direct extraction. Among the main sources of raw materials for the production of magnesium chloride, the most important is sea water. Magnesium can be directly precipitated from sea water in the form of magnesium hydroxide and converted to magnesium chloride [103]. Among the uses of magnesium chloride in the food industries are: coagulant; tofu production from soy milk; formula milk for babies, etc. The advantage of using magnesium chloride in food industry is that it can be a source of magnesium for the human body where it is one of the main minerals present quantitatively after potassium, calcium, phosphorus and sodium participating in carbohydrate, lipid metabolism, growth and cellular permeability. It is a catalytic element and also a plastic one and is a growth factor that helps regulate the balance of calcium in the human body having an anti-aging, anti-anaphylaxis and anti-atherosclerotic role [79]. To replace the nutrient losses in flour that occur during wheat processing and to reduce the risk of a deficiency in the body, the bakery industry has enriched white bread with various nutrients such as iron, thiamine, riboflavin and niacin [104]. Magnesium is considered to be one of the deficient elements in our diet although it is involved in the enzymatic reactions of carbohydrates, proteins, energy metabolism and maintaining the structural and functional integrity of human body tissues [105]. Along the benefits of magnesium consumptions are: fighting constipation, hypomagnesaemia, preventing convulsions in eclampsia/preeclampsia, preventing acute nephritis (pediatric and adolescent patients), cardiac arrhythmias secondary to hypomagnesaemia, etc. [106].

Salovaara [70] found that the addition of magnesium chloride in the dough reduces it development time and increases the water absorption capacity. Compared to the dough samples in which sodium chloride and potassium chloride were added in the bread recipe, it was found that there are significant differences from the rheological point of view only for the samples with magnesium chloride in dough recipe. According to this study, it was recommended the use of KCl as a substitute for NaCl even at high doses of 25–50% because this salt maintains the dough rheological properties, leading at the same time to a significant reduction of sodium in the finished products. This similar behavior between NaCl and KCl on the dough rheological properties is due to the fact that $K^+$ was classified equivalent to $Na^+$ in the lyotropic series which means that it would have similar abilities to cause protein aggregation and fortify the gluten network. It is considered a stabilizing cation that causes less protein hydration and stronger development of protein structure [99].

### 4.1.3. The Use of Potassium Chloride as a Substitute for Sodium Chloride in Bakery Products and the Benefits of Its Use on the Human Body

Potassium chloride (KCl) is a natural mineral salt obtained from salts from rocks and seawater, its extraction being similar to that of sodium chloride. Taking into account the dietary intake of potassium, it is clear that it has opposite effects to sodium consumption, namely a low risk of hypertension. While sodium intake is significantly increasing, the average overall potassium intake is below the WHO recommendations of at least 3510 mg of potassium per day.

Potassium is the most abundant cation in intracellular fluids and is an essential nutrient in maintaining cell functions, especially in excitable cells such as muscles and nerves. Being a major intracellular ion, it is manly found in foodstuffs that are obtained from living tissues. A higher amount of potassium was found in fruits and vegetables and less into meat products and cereals. In Western diets, food practices are not based on the fruits and vegetables consumption but rather on cereals and processed foods with a low content of nutrients which led to diets low in potassium and higher in sodium [107]. The best sources of potassium are fruits, vegetables, meat, fish, dairy and nuts. In starchy foods, potassium is found in higher amounts in whole meal flour and brown rice compared to rice and white flour. Milk, coffee, tea and other non-alcoholic beverages are among the main sources of potassium in the American adult's diets. According to the National Institutes of Health (NIH), the US estimates that the body absorbs about 85–90% of dietary potassium and different forms of potassium from fruits and vegetables including potassium phosphate, sulfate, citrate and others, but not potassium chloride (used in salt substitutes and some dietary supplements). Globally, mixtures of salts with potassium chloride are widely used for the partial or total substitution of sodium chloride. The use of potassium chloride can thus reduce the intake of sodium in food in a short period of time and to increase the potassium intake. There is an antagonism between the metabolism of potassium and sodium. Increasing the concentration of potassium in the human body leads to a decrease of the sodium concentration and to an increase in its elimination. At the same time, fluids are eliminated from the body. Diets high in potassium can help us to eliminate sodium from the body. Of all the types of salts used, potassium chloride is one of the most widely used as a substitute for sodium chloride because it has the best ability to transmit the salty perception of taste in food.

Therefore, the potassium chloride may be an interesting substitute for sodium chloride from the perspective of consumers, processors from the food industry but also from the point of view of consumer health. However, potassium chloride cannot be used in unlimited quantities because at high levels it loses its ability to give a salty taste in food and can often lead to a bitter, chemical and metallic taste.

Depending on the category of foodstuff in which it was introduced, potassium chloride has been used in different percentages to replace sodium chloride without affecting in a negative way the sensory characteristics of foods. For example, in aqueous solutions the taste of potassium chloride is perceived at a concentration of 20%, in food products such as pizza type good sensory properties was achieved up to a sodium chloride replacement with potassium chloride of a 25% level, in white and dark bread with 30%, in cheddar cheese with 46% and in feta cheese even up to 50% [66]. Studies have shown that this ingredient can have approximately the same technological functions in the dough making process and bread quality as sodium chloride leading to an improvement in the texture and shelf life of bread [67]. Doyle [52] suggested that the influence of KCl on dough rheological properties is similar to those obtained by using NaCl. Gengjun et al. [71] concluded that KCl could adjust the growth rate of the yeast cells, allowing the incorporation of more gas bubbles in the gluten network, which may have a positive effect on the rheological properties of wheat flour dough.

Consumption of potassium chloride and its use as a substitute for sodium chloride is safe for consumers health and is supported by the presence of potassium in a natural way in different foods. Therefore, the addition of potassium chloride to food has gained

regulatory acceptance in the US and the European Union. Experts recommend increasing potassium consumption by its addition in foods for the population because it has a low risk in terms of its adverse effects on consumers.

There is currently no upper limit on potassium intake at the global level, but based on estimates of current consumption in Europe, the European Food Safety Authority (EFSA) states that the risk of adverse effects on potassium intake from food sources per 5000–6000 mg/day is considered low for the clinically healthy population. Moreover, the long-term intake of potassium from supplements at a level of 3000 mg/day in addition to the consumption of foods containing potassium is also considered safe for the clinically healthy adult population [108].

Potassium chloride is one of the most common substitutes for sodium chloride in bread due to its ability to lead to a salty taste perception. It can replace sodium chloride up to a level of 30–40% without changing the characteristics of the finished product [5]. Although KCl is a possible option for reducing sodium in bakery products, a significant disadvantage of it is, as we mentioned before, the metallic taste conferred by this compound at high levels. That's way various studies have assessed the threshold of sensory acceptability. For example, Wyatt and Ronan [109] did not find significant differences between a control sample (with 100% NaCl) and other bread samples with 50%/50% (NaCl/KCl) bread, the highest scores in terms of acceptability having the NaCl ratio/KCl, 75%/25%. In contrast, Salovaara (1982) [70] found significant differences with a mixture of 60/40, but not with a mixture of 80/20. Replacement is not as critical factor for dark bread, and the use of a salt mixture in which the sodium content of the bread obtained is reduced by 32.3% and the K content is increased by 34.8% showed good results in terms of quality and taste of the bread obtained. Finally, in a systematic study involving the replacement of NaCl with $K^+$ salts, Braschi et al. [74] concluded that the best results, other than those for the control sample, were obtained with a 70/30 ratio of K-citrate or a 1:1 mixture of KCl and potassium bicarbonate; they also concluded the complete bioavailability of the incorporated potassium using these salts. Other possibilities that have developed include mixtures of commercial salt and salt from sources of low sodium and high potassium, calcium, or magnesium. The use of KCl in combination with Na glutamate or ribonucleotides may mask the bitter aftertaste. This is another interesting alternative, although this taste tends to be used only in the substitution of salt in meat products [110]. Potassium chloride (KCl) is usually the main choice and can be used to replace from 10% to 20% without major technological and sensory problems. Total KCl replacement is not recommended due to the unpleasant bitter and metallic taste, which limits consumer acceptability. For this reason, for a higher reduction of sodium in bakery products it is recommended to combine it with other food ingredients [12].

Therefore, the advantage of using KCl in the bread making recipe is a technological and healthy one due to the fact that it increases the potassium intake from the diet which is associated with a very low risk of hypertension, an opposite effect to sodium consumption [66]. A recent study estimated that potassium intake for the United States, Mexico, France, and the United Kingdom was 80%, 95%, 77%, and respectively 95%, under the recommendations of the World Health Organization on potassium intake [111].

### 4.2. The Use of Dry Sourdough as a Substitute for Sodium Chloride in Bakery Products

Another possibility of partial substitution of sodium chloride in bakery products is the use of sourdough. Sourdough is defined as a fermented semi-finished product obtained from flour and water in the presence of its own, natural microbiota and then dried in conditions to keep lactic bacteria in a viable state [112]. The industrial production of dry sourdough has over 40 years old and was initially used for obtaining products with a high acidity. Further, its main uses were to obtain bakery products with specific taste and aroma [113,114]. The advantage of using sourdough in baking is that it eliminates the leaven phase by shortening the time for bread making process. It allows obtaining the bread dough in a single phase leading to bakery products of a very high quality from the

technological and sensory point of view, similar to those obtained through dough making in a double or a triple phase. Nowadays, a wide range of sourdoughs are available on the market, which differs according to the flour used, as well as the specific flavor that exist on each sourdough. Sourdough in the form of an ingredient for bakery products corresponds to the new trend of clean labels, natural products, including a reduced use of additives [115]. The advantages of using dry sourdough in low-salt products to create healthy foods are evident. Belz et al. [24] suggested that the dry sourdough may compensate the effect of salt reduction on bread flavors and may lead to the good sensory characteristics of the finished products, such as crumb texture. Moreover, they also reported that the addition of sourdough in bread making, fermented with *Lactobacillus amylovorus*, for obtaining a low-salt bread, extended the shelf life compared to a control sample. Bread containing lactic acid bacteria (LAB) from fermented wheat germ had a saltier taste compared to a control bread. The salty taste was thought to be a combined effect of acidification and proteolysis. Due to the addition of a sourdough from rye malt fermented with glutamate that accumulated bacteria of the *Lactobacillus reuteri* species, it may be possible to reduce the salt content of bread from 1.5 to 1% (compared to flour), maintaining the taste and other characteristics of a standing quality. Sourdough improves the perception of salty taste and brings an additional intake of aromatic compounds. This is a useful functional ingredient for low-salt bread. Moreover, the use of dry sourdough is not limited to bread. One possibility is to incorporate sourdough into pastries or croissants to improve their flavor, texture and therefore palatability [12].

*4.3. The Use of Flavor Enhancers as Substitutes for Sodium Chloride in Bakery Products*

Flavor enhancers are compounds that do not have a salty taste, but they have the ability to intensify the salty taste of NaCl by activating receptors in the oral cavity. Some of them also mask the unpleasant taste of KCl. Yeast extracts are natural flavor enhancers, which the food industry commonly uses as substitutes for monosodium glutamate (MSG) and other artificial flavor enhancers. However, the effectiveness of MSG (as a salt substitute) is only partial, as it also contains sodium. Moreover, another problem is that the safety of MSG use is controversial because it has been associated with health problems (such as headaches, hyperactivity, and metabolic disorders) [13]. Glutamic acid-based combinations for salt replacement by flavor enhancement and intensification lead to good results [116]. As for yeast extract, although it is considered a natural and healthy alternative, it often contains MSG, which has been used in the food industry since the 1950s and has changed significantly in terms of quality, taste and functionality. Several low-sodium natural yeast extracts have been developed and can be used in a variety of salty foods. Yeast extracts are obtained from the water-soluble content of the cells, which contains concentrated amino acids, peptides, carbohydrates and mineral salts. In general, two types of extracts can be produced to the industrial level, using two different methods namely autolysis and hydrolyses. Autolysis of yeast extract is produced by processing yeast used in the bakery or beer industry. Cell walls are broken down by the use of heat or salt. This allows the enzymes present in the cell to break down the proteins and other cells compounds. The soluble compounds are then separated from the insoluble compounds and concentrated and pasteurized before being used in the food products [117]. The production of hydrolyzed yeast extracts implies the use of an acid, which starts the peptide bonds hydrolysis, releasing glutamic acid. Yeast extracts can be added to any salty food and are commonly used in sauces, spices and culinary products. Strong fleshy notes are also used to mask any unpleasant bitter taste resulting from the incorporation of potassium chloride into foods. This is a positive fact because the sensation of bitter taste induced by potassium chloride limits the amount in which it could be incorporated into food. This allows an increase in the amount of potassium chloride that could be added to foods containing yeast extracts in the bread making recipe [118]. It should also be mentioned that these ingredients do not have the same profile, in terms of NaCl functionality, on dough rheology, yeast fermentation rate, control of water activity and inhibition of microbial growth, which can create difficulties

in bakery industry. Therefore, even it presents a high potential for flavor improvement, yeast extracts are not well received by the consumers, which limits the acceptability of the food products where they are incorporated. To balance the overall flavor and overall characteristics of bread quality, the combination of salt substitutes with flavor enhancers is recommended to be used [12].

## 5. Perspectives

The strategies of bakery products reformulation in order to reduce salt are nowadays of a great interest to an international level, supported by the authorities and research community to benefit consumer health. It represents a great challenge for the bakery industry, since sodium chloride is one of the raw materials for bakery products with important effects on their technological and sensory properties. Due to the important role of NaCl it is difficult to reduce or to eliminate it completely from bakery products. However, baking industry undertakes efforts to reduce the salt content from baked products.

In the salt reduction strategies will be taken into account the economic and technical criteria, but also the acceptability of the bakery products with a low salt content by consumers.

Gradual reduction of salt from bakery products may be one of the best solutions for the industry and consumers in the future without significant costs. Moreover, the use of different salts with similar effects from the technological point of view (KCl, $MgCl_2$, $CaCl_2$, calcium lactate, calcium gluconate, etc.) to those of NaCl addition may play an important role of salt reduction in baked products but their uses are limited due to unpleasant taste. Some taste enhancers (amino acids, yeast extracts, glutamates, hydrolyzed vegetable proteins nucleotides, fermented sugars, etc.), B4 vitamin may be used in the future to the industrial level in combination with other salts to improve the salty taste of baked products.

To obtain bakery products with a low salt content the NaCl may be totally replaced with sea salt (a natural mix of NaCl, KCl, $MgCl_2$) with a low sodium content but it use may be limited due to it higher cost which may be problem in poor and developing countries in which bakery products are also low cost ones.

In order to produce bakery products with low salt amount, with technological and sensory characteristics comparable to bakery assortments obtained with a normal salt content, in the future it is recommended to continue researches to combine different ingredients with similar technological effects as those of NaCl which can improve each other their salty taste, to optimize production recipes and to implement different industrial processes such as encapsulation, inhomogeneous distribution of salt in the bakery process, etc.

Therefore, further researches are needed to obtain bakery products assortments with a low salt amounts, with good technological characteristics, by complying with WHO recommendation and consumers demand.

## 6. Conclusions

Salt is a minor component, which influences all phases from the bread making technological process, as well as the sensory properties of the bakery products obtained. From the technological point of view, salt presents a significant effect on dough mixing, fermentation and baking. Its affects dough development time, it presents a strengthening effect on the gluten network, makes it more extensible, it inhibits yeast activity leading to a decrease of the gas formed during the fermentation process, it extends the shelf life of the bakery products and it improves the bread sensory characteristics, especially taste, flavor, and the color of the crust. Due to the fact that World Health Organization recommends a daily sodium intake of 2 g equivalent to 5 g/day of salt intake food processors are concerned to reduce the salt content from their products. Bread is the most consumed food worldwide and that way it is the mainly contributor to the sodium daily intake.

The research made so far presents different strategies to reduce sodium from bakery products content (gradual reduction of salt levels from foods, uses of different types of mineral salts, of flavor enhancers, different ingredients with flavor compounds, etc.). Any

of these strategies of salt reduction are efficient in some ways. The easiest solution in salt reduction is it gradual reduction over time up to 25% from bakery products recipe. This does not imply any additional costs and technological changes. If this reduction is made under a longer period of time it does not affect the consumers acceptability. Salt replacers ($KCl$, $MgCl_2$, $CaCl_2$, calcium and magnesium salts as gluconate and lactate) may substitute in a limited way $NaCl$ from bakery products recipe due to their unpleasant flavor. However, they have technological advantages since their effect are similar to that obtained by $NaCl$ addition in bakery products. $KCl$ presents the most similar technological effects in baked products being the most common used salt for $NaCl$ substitution. Up to 20–30% of $NaCl$ substitution, $KCl$ may lead to good bakery products quality without any negative effects on its metallic and bitter taste. The use of sea salt with low sodium content it is in an increasing trend nowadays for it use in bakery products due to the fact that may conduct to good bakery products quality with similar characteristics with those obtained through $NaCl$ addition. Some taste enhancers (glutamates, yeast extracts, hydrolyzed vegetable proteins nucleotides, amino acids, fermented sugars, etc.) can also be used to intensify the salty taste perception in combination with different salts. These present an insignificant effect from the technological point of view and that way their combinations with other types of salts are necessary. The use of dry sourdough in bread recipe may also improve bakery products flavor and may be used in combination with different type of salts or taste enhancers to reduce the sodium content from bakery products.

The results indicate that different strategies may be used in obtaining bakery products with low sodium content of a good quality from a technological and sensory point of view, with benefic effects on consumer health and in agreement with WHO sodium recommendations values.

**Author Contributions:** A.V., A.D. and G.G.C. contributed equally to the study design, collection of data, development of the sampling, analyses, interpretation of results and preparation of the paper. All authors have read and agreed to the published version of the manuscript.

**Funding:** This research received no external funding.

**Conflicts of Interest:** The authors declare no conflict of interest.

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
