# Peer review of "Strategies for Reducing Sodium Intake in Bakery Products, a Review"

_applsci, doi:10.3390/app11073093_

Round 1

Reviewer 1 Report

Bread has been identified as one of important food sources of sodium in human nutrition. The manuscript reviews different strategies for reduction of sodium intake with bakery products, since the amount of dietary sodium needs to be limited. The review is quite comprehensive, however there is a lot of space for its improvement. Remarks and suggestions are bulleted below:

  • One of the important shortcomings of the manuscript is poor English, which in some parts affects the understanding of what is written. 
  • Line 234: The claims reported are defined by legislation, which is not referred. Please check and correct.
  • Lines 244-265: The data on salt content variability in breads and consequent amount of salt consumed with bread should also be critically assessed in a light of the amount of bread consumed in daily nutrition. Differences among countries in daily bread consumption are well established, in general it is well known which type of bread, which bread salt content presents the majority.
  • Line 433: Please check the last part of the sentence: "... a light crust, a weaker aroma and a tasteless taste." The term "tasteless taste" should be replaced by an appropriate one.
  • The manuscript is organized in chapters, however a systematic review would contribute to better systematization of the manuscript organization and may enable clear identification of data that would support the opinion of the authors stated in the conclusions (lines 876-882). In current form, this statement is not supported by data and needs to be rewritten.
  • Key findings on best strategies for sodium reduction in bakery products from the consumers', producers' and from political decision-makers' point of view are not drawn, which is one of important shortcomings of this manuscript. 

Author Response

23 March 2021

Dear Referee,  

We would like to thank the referee for the close reading and for the proper suggestions. We hope that we provide all the answers to the reviewer’s comments.

Thank you very much for the recommendations to publish our paper entitled “Strategies for reducing sodium intake in bakery products, a review”.

The present version of the paper has been revised according to the reviewer’s suggestions.             

We uploaded the corrected version of the article for which we used the red color/ purple color for the addition text.

GENERAL COMMENTS:

Referee comments: Bread has been identified as one of important food sources of sodium in human nutrition. The manuscript reviews different strategies for reduction of sodium intake with bakery products, since the amount of dietary sodium needs to be limited. The review is quite comprehensive, however there is a lot of space for its improvement. Remarks and suggestions are bulleted below:

Response: We would like to thank to the referee for its appreciations. We hope we made all the corrections that he/she requested in order to improve our manuscript.

Referee comments: One of the important shortcomings of the manuscript is poor English, which in some parts affects the understanding of what is written.

Response: We want to thank to the referee for the close reading of our manuscript. The article was corrected now by an Enghlish teacher.

Referee comments: Line 234: The claims reported are defined by legislation, which is not referred. Please check and correct.

Response: We made now a more cleary reference to the claims reported.

Referee comments: Lines 244-265: The data on salt content variability in breads and consequent amount of salt consumed with bread should also be critically assessed in a light of the amount of bread consumed in daily nutrition. Differences among countries in daily bread consumption are well established, in general it is well known which type of bread, which bread salt content presents the majority.

Response: We completely now more informations about salt variability in breads and we underlined in a more extensive way how salt is present depending on bakery products type, we offered and some examples of how it consumption are reflected in salt intake recommended by WHO.

Referee comments: Line 433: Please check the last part of the sentence: "... a light crust, a weaker aroma and a tasteless taste." The term "tasteless taste" should be replaced by an appropriate one.

Response: We revised.

Referee comments: The manuscript is organized in chapters, however a systematic review would contribute to better systematization of the manuscript organization and may enable clear identification of data that would support the opinion of the authors stated in the conclusions (lines 876-882). In current form, this statement is not supported by data and needs to be rewritten.

Response: We completed and  rewrote the conclusion part. We hope that now the conclusion part to be in a more proper way.

Referee comments: Key findings on best strategies for sodium reduction in bakery products from the consumers', producers' and from political decision-makers' point of view are not drawn, which is one of important shortcomings of this manuscript.

Response: We completly the manuscript with the perspectives part in which we hope to respond to the referee suggestions.

Reviewer 2 Report

The authors have discussed different strategies for reducing sodium intake in bakery products. In this regard, article is divided into different parts like introduction, physiological role of sodium chloride and legislative recommendations on its consumption, technological effect of sodium chloride in bread making and methods for sodium content reduction in bakery products. Although authors have discussed all important aspects, but this article is poorly drafted as far as English language is concerned. The detailed comments are given below and “Major Revisions” are suggested for this article to be published.

Methods for sodium content reduction in bakery products

  • The article needs significant improvement as far as English language is concerned. Here are few examples: Line 9 in abstract “levels in especially due to its high content…”, needs rectification as this sentence is not correct grammatically. The next sentence in same line states “This may causes…” can be modified as “This may cause…”. The sentence in lines 10-11 (Cereal products, and in particular bread, are the mainly source) should be modified as “Cereal products, and in particular bread, are the main Likewise, in line 13, replace “on it consumption” with “on its consumption”. The sentences from line 14-19 also need re-phrasing and grammatical corrections. The list goes on further, so it is suggested that authors should revise the language of this manuscript by consulting an English expert due to grammatical and phrasing errors.
  • The introduction part should follow the funnel approach by targeting the main elements of the topic rather than presenting general information. The authors are suggested to revise this part by focusing on various strategies for salt reduction in bakery products.
  • Addition of few figures will further improve the manuscript. For example, authors can illustrate the physiological process of sodium inside the body while discussing the understanding of various aspects of the effect of sodium chloride consumption on the human body.
  • Conclusion is more general and needs to be concise focusing on their main findings of review studies, conclusive remarks, and perspectives of various strategies and methods for reducing salt in bakery products.

Author Response

23 March 2021

Dear Referee,  

We would like to thank the referee for the close reading and for the proper suggestions. We hope that we provide all the answers to the reviewer’s comments.

Thank you very much for the recommendations to publish our paper entitled “Strategies for reducing sodium intake in bakery products, a review”.

The present version of the paper has been revised according to the reviewer’s suggestions.             

We uploaded the corrected version of the article for which we used the red color/purple color for the addition text.

GENERAL COMMENTS:

Referee comments: The authors have discussed different strategies for reducing sodium intake in bakery products. In this regard, article is divided into different parts like introduction, physiological role of sodium chloride and legislative recommendations on its consumption, technological effect of sodium chloride in bread making and methods for sodium content reduction in bakery products. Although authors have discussed all important aspects, but this article is poorly drafted as far as English language is concerned. The detailed comments are given below and “Major Revisions” are suggested for this article to be published.

Response: We would like to thank to the referee for its appreciations. We hope we made all the corrections that he/she requested in order to improve our manuscript. Regarding the English language all the manuscript was now corrected by an English teacher.

 Referee comments: The article needs significant improvement as far as English language is concerned. Here are few examples: Line 9 in abstract “levels in especially due to its high content…”, needs rectification as this sentence is not correct grammatically. The next sentence in same line states “This may causes…” can be modified as “This may cause…”. The sentence in lines 10-11 (Cereal products, and in particular bread, are the mainly source) should be modified as “Cereal products, and in particular bread, are the main Likewise, in line 13, replace “on it consumption” with “on its consumption”. The sentences from line 14-19 also need re-phrasing and grammatical corrections. The list goes on further, so it is suggested that authors should revise the language of this manuscript by consulting an English expert due to grammatical and phrasing errors.

Response: We would like to thank to the referee for his/her close reading of our manuscript. We corrected all the mentioned parahraphs and as we say all the manuscript was corrected from the English point of view.

Referee comments: The introduction part should follow the funnel approach by targeting the main elements of the topic rather than presenting general information. The authors are suggested to revise this part by focusing on various strategies for salt reduction in bakery products.

Response: We agree with the referee point of view. The introduction part too long and with many general informations. We deleted some parts of the introduction part, we completed more informations related to strategy part and also we made a figure related to reduction strategy which are the main part of this manuscript.

Referee comments: Addition of few figures will further improve the manuscript. For example, authors can illustrate the physiological process of sodium inside the body while discussing the understanding of various aspects of the effect of sodium chloride consumption on the human body.

Response: We want to thanks to the referee for his/her suggestions. We added 2 more figures in the manuscript now.

Referee comments: Conclusion is more general and needs to be concise focusing on their main findings of review studies, conclusive remarks, and perspectives of various strategies and methods for reducing salt in bakery products

Response: We agree with the referee point of view. We revised and complete the conclusions part and also we inserted in the manuscript the perspectives part.

Round 2

Reviewer 1 Report

Dear authors,

Thank you for providing revised version of your manuscript.

The manuscript has been revised and substantially improved. Nevertheless, there are still some inconsistencies that need to be addressed prior to publication. Please find them bulleted below:

  • Line 125: “… almost EU country …”. Please check the meaning of the sentence: a word (? every ?) is missing between words almost and EU.
  • Line 309: “… whereas in croissant, puff pastry and sweet the …”. Please check the meaning of the sentence: a word (a subject) is missing between words sweet and the.
  • Line 612: please check the numbering of figure. In this line Figure 5 is referred to, while a few lines further (line 621), the figure is numbered 4. Please check the text in line 612: “… exclusively to sodium ions (Na+) as it shown in Figure 5.” and in line 621: “Figure 4. Mechanism of salt perception”.
  • Lines 619-621: Please redo the Picture 4 as a reader might wrongly conclude, that receptors for salty taste are located only in fungiform papillae. Namely, the ability to taste e.g. sweet, salty, sour, bitter isn’t sectioned off to different parts of the tongue. The receptors that perceive these tastes are distributed all over the tongue and are not limited to a certain type of papilla. In this view, the box with “fungiform papillae” text and the type of tongue graphic are redundant / misleading. I suggest to reconsider the role of this figure, whether it substantially improves the quality of the manuscript and understanding of the content.

Author Response

25 March 2021

Dear Referee,  

We would like to thank the referee for the close reading and for the proper suggestions. We hope that we provide all the answers to the reviewer’s comments.

Thank you very much for the recommendations to publish our paper entitled “Strategies for reducing sodium intake in bakery products, a review”.

The present version of the paper has been revised according to the reviewer’s suggestions.             

GENERAL COMMENTS:

Referee comments: The manuscript has been revised and substantially improved. Nevertheless, there are still some inconsistencies that need to be addressed prior to publication. Please find them bulleted below:

Response: We would like to thank to the referee for its appreciations. We hope we made all the corrections that he/she requested in order to improve our manuscript.

Referee comments: Line 125: “… almost EU country …”. Please check the meaning of the sentence: a word (? every ?) is missing between words almost and EU.

Response: We want to thank to the referee for the close reading of our manuscript. The word every was missing from the sentence.

Referee comments: Line 309: “… whereas in croissant, puff pastry and sweet the …”. Please check the meaning of the sentence: a word (a subject) is missing between words sweet and the.

Response:  We want to thank to the referee for the close reading of our manuscript. We added the word bakery products after sweet.

Referee comments: Line 612: please check the numbering of figure. In this line Figure 5 is referred to, while a few lines further (line 621), the figure is numbered 4. Please check the text in line 612: “… exclusively to sodium ions (Na+) as it shown in Figure 5.” and in line 621: “Figure 4. Mechanism of salt perception”.

Response:  We want to thank to the referee for the close reading of our manuscript. We revised the number of the figures. However, we deleted the figure 4 from the manuscript according to the referee suggestions and therefore we deleted the reference to it in the manuscript.

Referee comments: Lines 619-621: Please redo the Picture 4 as a reader might wrongly conclude, that receptors for salty taste are located only in fungiform papillae. Namely, the ability to taste e.g. sweet, salty, sour, bitter isn’t sectioned off to different parts of the tongue. The receptors that perceive these tastes are distributed all over the tongue and are not limited to a certain type of papilla. In this view, the box with “fungiform papillae” text and the type of tongue graphic are redundant / misleading. I suggest to reconsider the role of this figure, whether it substantially improves the quality of the manuscript and understanding of the content.

Response: We discussed between us (the authors of this article) and we concluded that the referent is right and the figure 4 is not essential for the manuscript subject. The manuscript subject is about strategies for salt reduction in bakery products and not about salt perception of which we discussed only few phrases. Therefore we decided according to the referee suggestion to delete this figure from the manuscript.

Reviewer 2 Report

The authors have significantly improved the manuscript. The respective parts like introduction, general discussion and conclusion have been improved as per suggestions. The addition of figures provides better explanation of the content even for non-specialist readers. Further, authors have also revised the errors of English language as indicated previously. The article is recommended for publication.

Author Response

25 March 2021

Dear Referee,  

We would like to thank the referee for the close reading and for the proper suggestions. We hope that we provide all the answers to the reviewer’s comments.

Thank you very much for the recommendations to publish our paper entitled “Strategies for reducing sodium intake in bakery products, a review”.

The present version of the paper has been revised according to the reviewer’s suggestions.             

Referee comments: The authors have significantly improved the manuscript. The respective parts like introduction, general discussion and conclusion have been improved as per suggestions. The addition of figures provides better explanation of the content even for non-specialist readers. Further, authors have also revised the errors of English language as indicated previously. The article is recommended for publication.

Response: We would like to thank to the referee for its appreciations and for his/her recommendation for publish our paper.
